# Gestures as Scaffolding to Learn Vocabulary in a Foreign Language

**DOI:** 10.3390/brainsci13121712

**Published:** 2023-12-12

**Authors:** Ana Belén García-Gámez, Pedro Macizo

**Affiliations:** 1Mind, Brain and Behavior Research Center (CIMCYC), 18071 Granada, Spain; anabgarcia92@gmail.com; 2Departamento de Psicología Experimental, Facultad de Psicología, Campus de Cartuja, University of Granada, 18071 Granada, Spain

**Keywords:** foreign language learning, language learning strategies, iconic gestures

## Abstract

This paper investigates the influence of gestures on foreign language (FL) vocabulary learning. In this work, we first address the state of the art in the field and then delve into the research conducted in our lab (three experiments already published) in order to finally offer a unified theoretical interpretation of the role of gestures in FL vocabulary learning. In Experiments 1 and 2, we examined the impact of gestures on noun and verb learning. The results revealed that participants exhibited better learning outcomes when FL words were accompanied by congruent gestures compared to those from the no-gesture condition. Conversely, when meaningless or incongruent gestures were presented alongside new FL words, gestures had a detrimental effect on the learning process. Secondly, we addressed the question of whether or not individuals need to physically perform the gestures themselves to observe the effects of gestures on vocabulary learning (Experiment 3). Results indicated that congruent gestures improved FL word recall when learners only observed the instructor’s gestures (“see” group) and when they mimicked them (“do” group). Importantly, the adverse effect associated with incongruent gestures was reduced in the “do” compared to that in the “see” experimental group. These findings suggest that iconic gestures can serve as an effective tool for learning vocabulary in an FL, particularly when the gestures align with the meaning of the words. Furthermore, the active performance of gestures helps counteract the negative effects associated with inconsistencies between gestures and word meanings. Consequently, if a choice must be made, an FL learning strategy in which learners acquire words while making gestures congruent with their meaning would be highly desirable.

## 1. Introduction

Experimental science still has a lot of questions to solve in the fields of language learning and multilingualism. Due to the global and multicultural ambiance we are involved in nowadays, it is mandatory to be able to communicate in different languages. Early techniques for acquiring foreign language (FL) vocabulary employed a first language (L1)–FL word association strategy aimed at establishing connections between newly acquired FL words and their corresponding lexical translations in the native language [1,2,3,4]. To illustrate this, a native speaker of Spanish would learn that the English translation for “fresa” (L1) is strawberry (FL). Going a step further in the word association strategy, the keyword method [5] involves the utilization of a mnemonic method based on selecting an L1 word phonetically resembling a portion of an FL word (the keyword). In this approach, learners initially associate a keyword and the FL word verbally presented, followed by connecting the keyword to the L1 translation of the target word in the FL. For instance, the Spanish word “cordero” meaning “lamb” is associated with the word “cord” (phonetically related keyword). The effectiveness of these learning strategies has been confirmed in the initial states of FL acquisition, attributed to the formation of lexical associations between the learner’s L1 and the FL [6,7]. Nevertheless, when proficient bilingual individuals seek to express themselves in an FL, the most optimal processing route is the direct access to FL words from their associated concepts. The reliance on cross-linguistic lexical connections (L1-FL) and the retrieval of L1 words becomes superfluous when bilinguals communicate in an FL [8]. In addition, when learning programs based on the reinforcement of semantic connections are compared with lexically based learning procedures at the earliest stages of FL acquisition, advantages are found to be associated with conceptually mediated strategies [1,2,7,8,9,10,11].

In this context, understanding how fluent bilinguals process FLs can serve as a foundation for identifying learning methodologies that can foster this semantic processing pattern, thus constituting effective learning strategies [2,9,12,13,14]. These conceptually mediated strategies usually consist of protocols involving multimedia learning. The term “multimedia” encompasses the incorporation of five distinct types of stimuli within the learning protocol, which can be presented in combination or isolation: text, audio, image, animation and captions/subtitles [15,16,17]. When new information is presented in combination with different multimedia modalities, learners have the opportunity to organize and integrate it into long-term memory by forming multiple mental representations [18,19,20]. Notably, the picture association method demonstrates superior advantages compared to the word association method. Specifically, the method of FL words presented alongside images representing their meanings outperforms that of the presentation of FL words with translations in the L1 when behavioral measures are collected in single-word tasks [1,2,3,8,10,21], and tasks with words embedded in the context of sentences [3]. Moreover, electrophysiological measures are also sensitive to the benefit of the picture association method even after a single and brief learning session [7]. Likewise, the act of envisioning the meanings of the new vocabulary to be acquired has a facilitative effect on the learning process [22,23]. More relevant for the purpose of this manuscript, the integration of words and gestures has been proposed as a powerful FL learning tool, as these elements construct an integrated memory representation of the new word’s meaning [24,25,26,27,28,29].

### 1.1. Movements and Language

As humans, we possess the capacity to execute various types of gestures contingent upon the context or situation we encounter. In 1992, McNeill [27] proposed a simple gesture taxonomy considering many of the possible movements that we can make in a natural communication context. Within this classification, representational gestures encompass iconic gestures. Iconic gestures are employed to visually illustrate spoken content via the use of hand movements to represent tangible entities and/or actions. 

Across all spoken languages, Visual–manual communication frequently complements speech in everyday communicative interactions [30,31]. These co-speech gestures are often and naturally integrated into communication, involving the combination of facial and body movements, including hand gestures, alongside spoken language. In this sense, sounds and movements act as a means to integrate information that enhances the process of communication [32]. In fact, a number of studies have sustained the significance of movement in language processing [33,34]. As an interesting example, Glenberg and colleagues [35] observed a correlation between language comprehension and the direction of movement performance. Based on a container’s position, participants were tasked with transferring 600 beans (individually) from a larger container to a narrower one either toward or away from their bodies. Subsequently, sentences describing movements, both meaningful and meaningless, were presented to make a plausibility judgment. The results revealed that the processing time for sentences depended on whether or not the bean’s movement direction corresponded with the sentence’s description (either moving away from or towards the body). Consequently, the execution of actions had an impact on language comprehension. Notably, exploring brain activation in the presence of multisensory input, the integration of sounds from spoken language and accompanying gestures shows selective activity in the left posterior superior temporal brain areas and the right auditory cortex [36].

Numerous theoretical frameworks exist to elucidate the associations between speech and gestures. These accounts primarily delve into the underlying representations involved in gesture processing. These models can be differentiated by considering the interplay of visuospatial and linguistic information. Various perspectives such as the sketch model [37], the interface model [38] and the gestures-as-simulated-action (GSA) framework [39] propose that representations of gestures are rooted in visuospatial images and highlight the connection between gestures and visuospatial imagery. Conversely, other models emphasize the close interrelation between gesture representations and linguistic information, such as the interface model [38] and the growth point theory [27,28].

Another distinction among these models lies in how gestures and speech are processed. Some models propose that gestures and speech are processed independently, interacting when forming communicative intentions, or during the conceptualization phase to facilitate effective communication. This is evident in models like the lexical gesture process model [40], the sketch model [37] and the interface model [38]. In contrast, other models posit that gestures and speech function collaboratively within a single system. This perspective is reflected in models like the growth point theory [27,28] and the GSA framework [39]. For instance, the gesture-in-learning-and-development model proposed by Goldin-Meadow [41,42] suggests that children process gestures and speech autonomously, and these elements integrate into a unified system in proficient speakers [43,44].

In conclusion, it is reasonable to infer that the gestures we utilize while attempting to convey a concept play a role in both language production and comprehension, ultimately enhancing the overall communication process [45,46,47].

### 1.2. Gestures as FL Learning Tool

Several studies have underscored the significance of different types of gestures in FL learning (e.g., [47,48,49,50,51,52], for reviews). Generally, it is broadly accepted that gestures have a beneficial impact on word learning, encouraging their integration into FL instruction, aligned with a natural language teaching approach (see [53,54]; however, see [24,55], which supports the reduced effect of gestures in segmental phonology acquisition). More relevant to the purpose of this paper, past studies have addressed the importance of iconic gestures in language comprehension [56] (for a comprehensive review, see [57]), as well as in language production (see [58], for a review on gestures in speech).

Three primary accounts provide explanations of the advantageous impact of iconic gestures in learning vocabulary in an FL.

The *self-involvement explanation* posits that gestures promote participant engagement in the learning task, enhancing attention and favoring FL vocabulary acquisition [59]. This increase in attention is primarily attributed to heightened attentional and perceptual processing due to the execution of gestures or the use of objects to recreate actions [60]. In this context, it is important to note that the motor component itself might not be the primary cause of improvement [61]. Rather than solely the gesture itself, it is the multisensory information it conveys that leads to increased attention and improved semantic processing [62]. Therefore, according to this perspective, learning new FL words accompanied by gestures promotes vocabulary acquisition, irrespective of whether or not a gesture is commonly produced within a language or represents the same meaning as the word to be learned [63,64]. To illustrate this, if the learner needs to acquire the word “teclear” in Spanish whose translation in English is “to type”, the mere fact of performing a movement associated with the new word would facilitate the process independently of any other intrinsic characteristic of the gesture.

The *motor trace perspective* argues that the physical aspect of gestures is stored in memory, creating a motor trace that assists in the process of acquiring new words in an FL [65,66]. According to this viewpoint, physical enactment is crucial because it enables the formation of a motor trace linked to the word’s meaning. Recent neuroscientific studies, using techniques such as repetitive transcranial magnetic stimulation, provide support for the involvement of the motor cortex in written word comprehension [67]. Additionally, evidence suggests that familiar gestures might engage procedural memory due to their reliance on well-defined motor programs [68]. Consequently, the interplay of procedural and declarative memory could enhance vocabulary learning. Hence, well-practiced familiar gestures promote FL learning to a greater extent than unfamiliar gestures do (e.g., the gesture of typing on a keyboard vs. touching the right and the left cheeks with the right forefinger sequentially). However, this perspective supports the idea that the impact of gestures operates regardless of their meaning and well-practiced gestures might benefit learning regardless of whether or not they match the new word’s meaning.

The *motor imagery perspective* suggests that gestures are tied to motor images that contribute to a word’s meaning [69]. Specifically, executing a gesture while processing new words fosters the formation of a visual image linked to the word’s conceptual information, enhancing its semantic content [49,70]. Functional connectivity analyses provide neurobiological evidence suggesting that the hippocampal system plays a role in linking words and visual representations [71]. In this manner, the facilitation effect of gestures is heightened when they align with the meaning of the words being learned, compared to cases where gestures and word meanings do not match. This constitutes the primary point of disagreement with the motor trace theory. Additionally, this perspective points out that learning words with gestures of incongruent meanings can lead to semantic interference and reduced recall due to the creation of a dual task scenario, which ultimately has an adverse impact on the learning process [72,73].

From our view, these three perspectives are not mutually exclusive, but rather highlight different aspects of gestures’ effect on FL learning. A gesture accompanying a word could increase self-involvement (gestures enhancing attention to FL learning), create a motor trace (meaningful movements) and/or evoke a semantic visual image integrated with the word’s meaning.

### 1.3. Empirical Evidence Regarding the Role of Gestures in FL Learning

Empirical evidence pertaining to the role of movements in FL instruction indicates that enhanced vocabulary learning outcomes are observed when learners acquire FL words accompanied by gestures that illustrate the practical use of objects whose names they are required to learn [74,75,76]. Many years ago, Asher [77] was a pioneer in introducing movements in the FL learning process. He presented the Total Physical Response strategy as an effective means of acquiring vocabulary. This strategy involved a guided approach where students received instructions in the target language (FL). For example, children were taught the Japanese word “tobe” (whose meaning is ‘to jump’ in English), and each time they were presented with this word, they physically executed the corresponding movement (to jump). The author observed an advantageous impact linked with the integration of gestures in FL word instruction, an effect that has been confirmed across various educational domains [78,79] (although see [80] for an alternative view).

Then, Quinn-Allen [81] conducted the first empirical study that delved into the influence of iconic gestures in FL acquisition. In this study, English speakers, when presented with French expressions (e.g., il est très collet monté?—he takes himself seriously), achieved better results when they were exposed to and reproduced a symbolic gesture illustrating the sentence’s meaning (e.g., head up, one hand in front of the neck, and the other hand lower as if adjusting a tie), compared to a control group in which no gestures were presented. The efficacy of gestures observed in Quinn-Allen’s study aligns with the postulates of the three theoretical perspectives, described in the previous section, because only a congruent gesture and a no gesture condition were introduced. This work laid the foundation to study the effect of gestures in FL acquisition; however, the causes underlying this improvement associated with the use of gestures during FL learning remain unclear. Further experimental work has answered this question by including several gesture conditions and manipulating the correspondence between the gestures and new words’ meaning [49,70,75,82].

Another benchmark study in the field is the work by Macedonia and collaborators [70]. In this study, German speakers learned new words in an artificial language created by the authors (Vimmi) that served as an FL. The new nouns were presented in combination with either meaningful congruent iconic gestures (such as the term “suitcase” paired with a gesture of lifting an imaginary suitcase), or meaningless gestures (such as the word “suitcase” accompanied by a gesture involving touching one’s own head). The results revealed that new words associated with iconic gestures had better recall compared to those coupled with meaningless gestures. These results suggest that gestures introduce something else beyond merely involving the participant in the task. The simple engagement associated with the gesture’s performance cannot explain the advantages found in the iconic gestures condition. However, in this case, both the motor imagery and motor trace theories potentially explain the heightened recall observed in the context of iconic gestures relative to meaningless gestures. Iconic gestures might enhance FL learning due to their semantic richness and their higher frequency of use compared to that of meaningless gestures, resulting in stronger motor activation.

Other researchers have employed additional experimental manipulations to distinguish between explanations rooted in the motor imagery account and those originating from the motor component of gestures. Studies involving monolingual speakers have investigated congruity–incongruity effects by intentionally misaligning the semantics of words and the meanings conveyed via gestures [74,83,84] (see [85] for congruity effects in an unfamiliar language context). Kelly and colleagues [48,86] employed an event-related potential study alongside a Stroop-like paradigm. New words (e.g., “cut”) and corresponding gestures were presented to participants. Words and gestures could be either semantically congruent (e.g., a cutting movement) or incongruent (a drinking movement). This study revealed an attenuated N400 response to words paired with congruent gestures compared to incongruent ones, thus displaying a semantic integration effect [87]. These results suggest that gestures are incorporated within new words’ meanings, producing benefits when gestures and words’ meanings match, and producing interference when learners perceive a conceptual mismatch. In this case, the gestures used in both conditions, congruent and incongruent, were familiar to participants as there was an equal level of engagement. Hence, the results obtained in this study could not be explained via the self-involvement or the motor trace accounts. The motor imagery theory would be unique in pointing out differences between the use of congruent and incongruent iconic gestures that rely on the meaning match or mismatch between gestures and words, respectively.

Taken together, previous studies have confirmed the positive effect associated with the use of congruent gestures on FL vocabulary learning. However, considering the experimental conditions included in past studies, there is a lack of empirical evidence comparing the consequences of several conditions in a single study within a single participant’s sample. For example, in different studies, the meaningless gesture condition is not included in the experimental design. In this context, it is not possible to determine the degree to which the congruent and incongruent conditions produce a facilitation or an interference effect, respectively, due to the lack of a baseline condition (see [88] for a review of a comparable experimental paradigm concerning Stroop tasks). We addressed this concern in our lab by developing Experiments 1 and 2.

### 1.4. Effects of “Seeing” and “Acting” Gestures While Learning an FL

In the realm of education, the potential advantages of learning through actions rather than solely through observation have been a subject of debate for decades [89]. The perspective of “learning-by-doing” advocates for active individual engagement in the learning process via the execution of actions while learning is taking place. Learning by doing can have a positive influence on the formation of neural networks underlying knowledge acquisition and the performance of cognitive skills [90]. This positive effect has been verified across various instructional domains such as language acquisition, learning through play, new technology utilization and online courses [78,79,91]. However, as reported in the next paragraphs, empirical evidence is not fully consistent in terms of the advantages of self-generated movements compared to those of the mere observation of actions performed by others. 

Within this context, various studies have explored the divergences observed when participants reproduce the experimental tasks by themselves or when they merely observe the experimenter [92,93]. Previous studies have pointed out that self-generated movements enhance cognitive processing. Out of the linguistic context, Goldin-Meadow and colleagues [90] directly compared the effect of the generation and mere observation of gestures. In this study, children participated in a mental transformation task, determining whether or not two forms presented in different orientations constituted the same figure. This task was selected due to its direct link between mental rotation and motor processing. When mentally rotating a target, premotor areas involved in action planning become active [35,94], and participants naturally and spontaneously used gestures when explaining how they solved the task [95]. Goldin-Meadow and colleagues [90] showed that children attained better results when they were instructed to physically perform the rotation necessary to solve the transformation task, rather than when they were merely observing the experimenter’s movements.

If we move to the field of language learning, additional empirical studies have emphasized the significance of self-generated movements in acquiring linguistic material. In 2011, James and Swain [96] planned a study in which children were taught action words associated with tangible toys. Children who manipulated the objects during learning exhibited activation in motor brain areas when hearing the words they had learned. Thus, involvement in motor actions enriches the acquisition of new words, potentially attributable to the formation of a motor trace activated during subsequent information retrieval. Similarly, Engelkamp and collaborators [97] showed that at a higher level of linguistic processing, sentence recall was better when participants were actively involved in performing the actions during the learning phase as opposed to a condition where they solely listened and memorized the material.

On the other hand, Stefan and colleagues [98] reported activity in the motor cortex while participants observed movements (e.g., repetitive finger movements), resulting in a memory trace that resembled the activation pattern during motor action performance. If there is an activation overlap between movement observation and performance, is the advantage associated with generating movements that clear? In fact, previous research has found confounding results or similar outcomes when participants engaged in producing actions or when they solely observed actions performed by others [99]. At the lowest level of linguistic processing, the production versus the mere observation of hand gestures has a limited impact on learning segmental phonology or phonetic distinctions in an FL [24,55]. In more advanced linguistic stages, where hand gestures have demonstrated a positive influence on learning, various studies show that self-generated movements and the observation of gestures yield comparable outcomes [100]. In the specific context of FL acquisition, Baills and collaborators [101] found similar results during the learning of Chinese tones and words when pitch gestures (metaphoric gestures mimicking speech prosody) were used. Recently, young-adult English speakers were acoustically presented with Japanese verbs while an instructor performed iconic gestures. Comparable outcomes were observed when participants learned the words solely by observing the instructor’s gestures or by mimicking her movements [102]. In a midway position, Glenberg and colleagues ([33], Experiment 3) investigated the impact of movements on sentence reading comprehension in children and found intermediate outcomes. Children were presented with narratives set in a scenario (e.g., a farm) involving several elements (e.g., a sheep or a tractor). One group manipulated the objects mentioned in the text, while another group imagined carrying this out. Children who physically manipulated objects achieved better results, while in the imagined condition, children showed only a modest improvement compared to a read-only condition.

In summary, the self-generation of movements during learning appears to yield positive effects in both non-linguistic tasks [90] and FL instruction [10,97,103]. The creation of a more comprehensive semantic representation in memory, encompassing both verbal and motor information, facilitates greater accessibility to previously acquired knowledge. This underlies the advantageous impact of performing movements while learning is taking place [96,97]. Nevertheless, other studies propose that merely observing gestures is sufficient for learning, regardless of whether participants engage in the gestures themselves or not [102]. This conflicting pattern of outcomes may stem from methodological differences between studies, such as participant demographics, including children [90] versus undergraduate students [55], and the nature of the learning tasks, ranging from dialogic tasks [15] to segmental phonology [24]. Notably, many studies supporting the positive effects of self-generated gestures involve semantically rich materials like words [103] or sentences [97], while some other studies indicating no differences between gesture observation and production focus on non-semantic linguistic levels (e.g., segmental phonology, [24,55]) or the manipulation of the gesture conditions is conducted in different experiments [101]. An additional purpose of our experimental series was to address these aspects comprehensively, investigating the performance versus observation of gestures in different groups while participants learn words accompanied by iconic gestures conveying semantic information.

### 1.5. Gestures in Verbs and Nouns Learning

When considering studies on FL vocabulary learning, there are many theoretical and practical issues that must be attended. In the case of learning and gestures, the close relationship between movements and verbs has a special role regarding codification and recall processes. The GSA framework establishes specific predictions about the role of gestures on learning different types of words. This theory posits that gestures emerge from simulated actions and perceptions, which underlie mental imagery and language production [39]. While viewing the shape or size of an object (nouns) does involve simulated movements, verbs and motor actions are more directly connected at a semantic level. Consequently, gestures would exert a more significant influence on verb learning compared to noun learning. In fact, it has been proposed that noun and verb lexical acquisition mechanisms might implement a bipolar approach. In this way, the cognitive mechanisms for nouns and verbs acquisition would be different [103].

In a study conducted by Hadley and colleagues [104] involving preschool children, the impact of gestures on teaching different word types was directly examined. Results revealed that while concrete nouns exhibited higher learning rates, employing gestures during verb instruction acted as a scaffold for the accompanying verbal contents. This insight elucidates why the majority of research evaluating gesture effects in FL learning has employed verbs as instructional material [47,75,85]. Many verbs (e.g., those describing actions involving manipulable objects) closely correlate with movements [82]. In fact, prior studies confirm that the semantic representation of verbs inherently includes a gestural or motor component [47,105,106,107].

However, beyond gestures, it is commonly found that verbs are more difficult to acquire compared to noun learning. Concrete nouns possess distinct perceptual attributes that facilitate the acquisition of new words, whereas verbs convey dynamic information that helps in the retrieval of the motion meaning [108]. Numerous studies have demonstrated that children acquired English nouns more effortlessly than verbs in a natural context [104,109]. However, this phenomenon seems to be culturally specific and although it appears in English-speaking cultures, the effect is blurred in non-English-speaking cultures [109,110]. One possible explanation for these cross-linguistic differences could be the particular emphasis that English speakers place on nouns when interacting with children during the acquisition of their native language. 

As far as we know, the study by García-Gámez and Macizo in 2019 [111] was the first example of research that directly compared nouns and verbs in the context of adult individuals learning FL words with accompanying gestures. The results obtained in this study will be presented and discussed in the current review paper.

### 1.6. The Current Series of Studies

Like many of the studies described below, we aimed to shed light on strategies that could help learners on the road to FL vocabulary acquisition. Taking into account all the information provided in Section 1, we designed a series of three experimental studies centered on contributing to the existing literature in the field.

Firstly, we wanted to investigate which of the theories explaining the role of iconic gestures in FL learning would align more effectively with the observed learning results. To determine this, we created different learning conditions by manipulating the semantic relation between gestures and words (congruent, incongruent, meaningless and no gesture). A more detailed description of the learning conditions is provided in the methods section.

In Experiments 1 and 2, we incorporated the different learning conditions and included nouns and verbs, respectively, as learning material. Although nouns have previously proven to be faster and easier to acquire, the inherent semantic association between gestures and verbs could contribute to enhancing action verb acquisition.

Finally, in Experiment 3, we also added the different learning conditions depending on the word–gesture semantic relation. Additionally, we wanted to investigate if the physical engagement of the learners could contribute to word learning beyond the mere observation of gestures performed by an instructor. To achieve this, two experimental groups were introduced to learning strategies involving word acquisition through either gesture imitation or observation.

## 2. Review of Research from Our Lab: Experiments 1 and 2

In Experiments 1 and 2, we assessed various perspectives that elucidate the potential role of gestures in FL vocabulary learning. Furthermore, Experiment 1 utilized nouns as the learning material, while in Experiment 2, verbs were employed as the words to be acquired. The remaining experimental conditions were maintained across experiments.

### 2.1. Participants

First of all, regarding the type of population, we selected Spanish “monolingual” speakers. All the participants were young-adult students from the University of Granada that received course credits as reward for participating. Nowadays, it is confusing to classify a Spanish person as monolingual. As is often the case in other countries, the younger generations are exposed to FL learning in regular education. In this context, we decided to recruit participants who were as minimally proficient in any FL as possible. To achieve this, we established the following inclusion criteria regarding the participants FL proficiency level. On a daily basis, they needed to confirm the following:
They had no contact with any language other than Spanish, whether spoken or in sign language.Their most recent exposure to an FL had to occur during high school.They had never received any formal instruction in an FL beyond regular education.They had never obtained a certification in any FL.

The language proficiency level of the participants was relevant for the experimental design in these studies. Previous research has shown that there are differences in the way new languages are acquired depending on how proficient speakers are in other languages [112,113]. For instance, learning a third language (L3) is generally assumed to be less demanding or costly than learning a L2 is [114,115]. The existing literature suggests that bilingual experience confers individuals with tools that facilitate the L3 learning process (for a comprehensive review, see [116]).

All participants provided written informed consent before engaging in the experiment. Participants were required to report no history of language disabilities, and they all needed to have either normal visual acuity or corrected-to-normal visual acuity. The data obtained in the studies were treated anonymously by the researchers.

### 2.2. Experimental Conditions

The studies involved 4 FL vocabulary learning conditions that were manipulated within participants as follows. In Figure 1, we show an example of the experimental design for each learning condition:(a)*Congruent condition*: Gestures reflecting movements commonly performed when manipulating a concrete object were paired with L1–FL word pairs. For example, “teclado (keyboard in Spanish)-saluzafo (Vimmi translation)” was coupled with the gesture of typing with both hands fingers as if we had a keyboard in front of us.(b)*Incongruent condition*: A meaning mismatch was introduced between the L1 word and the semantic content of the gesture. For instance, “teclado-saluzafo” was paired with the gesture of striking something with a hammer.(c)*Meaningless condition*: L1–FL word pairs were paired with unfamiliar gestures. For instance, “teclado-saluzafo” was accompanied by a gesture of touching the forehead and then one ear with the right forefinger.(d)*No gesture condition*: Participants had to learn Spanish (L1)–Vimmi (FL) word pairs without the use of gestures. For example, they had to associate “teclado” with “saluzafo”.

**Figure 1 brainsci-13-01712-f001:**
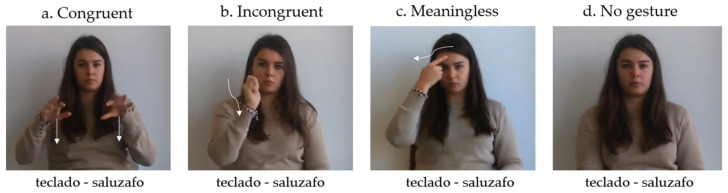
Learning conditions implemented in this series of studies. Different gesture conditions were presented along with Spanish (L1)–Vimmi (FL) translations (nouns in this example). In the example, the pair *teclado* (“keyboard” in L1)–*saluzafo* (FL translation) was accompanied by (**a**) the gesture of moving typing with both hands fingers as if we had a keyboard in front of us (congruent condition); (**b**) the gesture of striking something with a hammer (incongruent condition); (**c**) the gesture of touching the forehead and then one ear with the right forefinger (meaningless condition); (**d**) no gesture (no gesture condition).

### 2.3. Materials

Spanish words were selected to act as the L1. In Experiment 1, 40 concrete nouns denoting objects were selected as the learning material (e.g., spoon, comb, etc.). In Experiment 2, 40 Spanish verbs were used to be coupled with the FL translations. Linguistic variables such as the number of graphemes, phonemes and syllables, lexical frequency, familiarity or concreteness were carefully controlled.

As an FL to be learned, we selected an artificial language already developed, called Vimmi [49,70]. The corpus of Vimmi words has been design to eliminate factors that could potentially bias the learning of particular items. This includes avoiding patterns like the co-occurrence of syllables and any resemblance to words from romance languages such as Spanish, English and French. Vimmi words were meticulously chosen to be pseudowords in the L1 of the participants (Spanish), thus maintaining proper Spanish phonology and orthography but without semantic content (see [117], for a debate considering these factors in vocabulary acquisition). As reported by Bartolotti and Marian [118], divergent linguistic structures across languages can hinder new vocabulary acquisition. Vimmi matches the orthotactic probabilities of Latin languages such as Spanish, French or Italian. When the set of Vimmi words was selected in our studies, different linguistic variables were controlled such as number of graphemes, phonemes and syllables.

Once the Spanish and Vimmi words to be learned were selected, both sets of words were randomly paired. This resulted in 40 word pairs in total, with each pair consisting of an L1–Spanish word and an FL–Vimmi word. These 40 word pairs were then randomly divided into four conditions, each containing 10 word pairs. Each set of 10 pairs was associated with one of the learning conditions. To ensure a balanced distribution of gesture conditions across the word sets, four lists of materials were created. Hence, across the lists, all 40 word pairs were evenly distributed over the four learning conditions, ensuring that, across participants, every word pair appeared in the four learning conditions. Spanish nouns and verbs across the four sets of word pairs were equated with lexical variables. There were no significant differences across the L1 word sets in terms of number of graphemes, number of phonemes, number of syllables, lexical frequency, familiarity or concreteness. Likewise, Vimmi words in the four sets were equivalent in terms of number of graphemes, number of phonemes and number of syllables. Furthermore, we controlled the L1–FL word similarity by calculating the number of shared phonemes between words.

The gestures used in the studies included only hand movements. We implemented iconic gestures depicting common actions people normally perform when interacting with objects (e.g., mimicking writing a letter or brushing hair) [27,119]. The videos showing the hand gestures were recorded by the first author of this paper. These gestures were used in the congruent and incongruent gesture conditions. Furthermore, in the meaningless condition, gestures consisted of small hand movements without iconic or metaphoric connections to the meanings of the accompanying word (e.g., forming a fist with one hand and raising the fingers of the other hand). Care was taken to ensure that these meaningless gestures shared visual characteristics with meaningful gestures, such as spatial disposition and simple hand movements trajectory. In this condition, 10 different movements were selected, and all participants were exposed to the same set of gestures.

Furthermore, we aimed to ensure that the different gesture conditions varied in the extent to which the semantics of the word corresponded to the meaning of the accompanying gesture. To achieve this, a group of participants that did not perform the main experiment participated in a pilot study. A video displaying a gesture (without sound) appeared at the top of the screen, while a Spanish word was shown at the bottom. Participants were tasked with assessing the correspondence between the word’s meaning and the gesture by rating it on a scale ranging from 1 to 9 (indicating high mismatch or strong match, respectively). The results of the pilot study showed differences between nouns and verbs included in the three gesture conditions. Specifically, in the congruent condition, the gesture–word pairs received significantly higher ratings compared to those from both the meaningless and incongruent conditions. Furthermore, there were differences between the incongruent and the meaningless conditions. Thus, the three conditions with gestures used in our series of studies differed in terms of the association between the words’ and gestures’ meanings.

### 2.4. Procedure

Participants were exposed to 3 learning and concurrent evaluation sessions that took place on 3 consecutive days with a delay of 24 h between sessions. In each session, learning and evaluation phases were separated by a 15 min break. The experimental software used for stimuli presentation and data acquisition was E-prime 2.0 [120]. The experimenter informed participants that the sessions could be video-recorded to ensure they followed the instructions provided. A video camera was installed for this purpose although no recording was actually carried out. This experimental procedure was approved by the Ethical Committee on Human Research at the University of Granada (Spain) associated with the research project (Grant PSI2016-75250-P; number issued by the Ethical Committee: 86/CEIH/2015, approval date: 31 July 2015) awarded to Pedro Macizo. It was conducted in accordance with the 1964 Helsinki Declaration and its subsequent amendments.

#### 2.4.1. Vocabulary Learning Phase

As previously mentioned, we implemented some variations to adapt the procedure to the goal of each study. Experiment 1 included nouns as learning material while in Experiments 2, verbs served as words to be learned.

The learning phase lasted approximately 1 h per session. The stimulus presentation procedure was structured by experimental conditions (blocked stimulus presentation per learning condition) [70]. This blocked design was adopted to minimize the cognitive effort associated with constantly switching between conditions where participants had to perform gestures and those without gestures. A single learning block contained the 40 L1–FL pairs (10 from each learning condition). Each word was presented 12 times resulting in each participant receiving 12 blocks and hence 480 trials. Short breaks were incorporated between the learning blocks. The word pair presentation was randomized within each condition, and the order of learning conditions within a block was also counterbalanced. 

The experimenter video-recorded the gestures, which varied in nature (being congruent, incongruent or meaningless) depending on the accompanying word and consequently, the learning condition. Each recorded gesture was repeated twice and lasted a duration of 5 s. The video appeared in the middle top part of the screen. In addition, in all experimental conditions, participants were presented with a Spanish–Vimmi (L1–FL) word pair visually displayed at the bottom of the screen. These word pairs were presented with the experimenter in a static position (no gesture condition) or accompanied by a gesture (remaining conditions) (see Figure 1). 

The participants received instructions to read and repeat aloud each L1–FL word pair two times. In the conditions involving gestures, participants were additionally tasked with performing the corresponding gesture alongside each word pair during vocalization. They were required to synchronize the initiation of the gestures and speech production and each word pair and accompanying gesture was repeated twice. As an example, when the word pair “teclado-saluzafo” appeared along with the congruent gesture (Figure 1a), they were instructed to read the word pair while simultaneously performing the corresponding gesture of moving fingers as if typing on a keyboard. After producing the word pair twice while also mimicking the experimenter’s movements twice, participants pressed the space bar to proceed to the next trial.

#### 2.4.2. Evaluation Phase

The recall of Vimmi words was evaluated by implementing two tests: Spanish words were presented to be translated into Vimmi (forward translation from L1 to FL) and Vimmi words were presented to be translated into Spanish (backward translation from FL to L1). These tasks have previously proven to be an efficient measure of vocabulary acquisition [11,121]. To mitigate potential order or practice effects, the sequence of the translation tests was randomized across the three training sessions and among participants. Each translation task comprised 40 Spanish words and their corresponding 40 Vimmi translations for both forward and backward translations. Each word remained on the screen until participants provided their response. Recorded oral translations were later analyzed for accuracy, and response times (RTs) were measured from word presentation to the start of oral translation. On average, the learning assessment took about 10 min, varying based on individual performance.

### 2.5. Predictions and Hypothesis

When considering the theoretical perspectives of the role of gestures in FL vocabulary acquisition (previously described), namely the self-involvement account [59], the motor trace account [65,66] and the motor imagery account [69], it becomes possible to formulate specific predictions attending to our experimental conditions (see Figure 2 to observe the predictions of each of the theories for the learning conditions included in our studies). For a more comprehensive understanding of these predictions, a detailed description of the theoretical frameworks is available in the “Gestures as FL Learning Tool” section of the introduction.
If gestures primarily serve to enhance the participant’s engagement in the learning tasks, then all conditions involving gestures should lead to improved FL vocabulary acquisition compared to that in the no-gesture condition.If the motor trace left by gestures aids in learning new vocabulary, gestures which are familiar to participants (those in the congruent and incongruent conditions) should be associated with better FL vocabulary learning outcomes than those from less familiar gestures (meaningless condition) and the condition without gestures.Furthermore, the motor imagery account suggests that learning meaningful gestures could either facilitate or interfere with vocabulary acquisition, depending on the alignment between the gesture and word meanings. Thus, congruent gestures may enhance vocabulary acquisition, while incongruent gestures would make it harder to learn new words. Conversely, it can be the case that meaningless gestures could become distinctive and facilitate the encoding of new word [122].

**Figure 2 brainsci-13-01712-f002:**
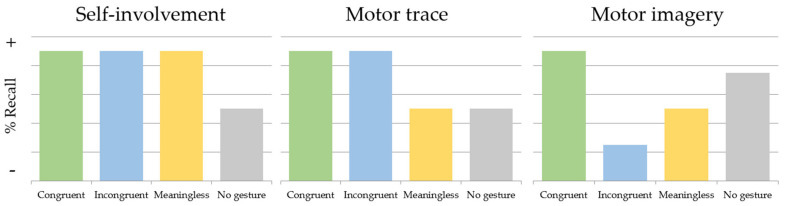
Visual representation of the predictions established through each theory on the role of gestures in FL vocabulary acquisition (self-involvement, motor trace and motor imagery) for each learning condition (congruent, incongruent, meaningless and no gesture).

As previously mentioned, it is worth noting that these perspectives are not mutually exclusive and different explanation may act together to form a comprehensive understanding of how gestures contribute to FL learning.

In Experiments 1 and 2, we established a direct comparison between noun and verb learning. The prevailing notion is that verbs might be more difficult to acquire compared to nouns [106,110]. Additionally, it is commonly accepted that action verbs inherently incorporate a gestural or motor component within their mental representation [105,107]. Therefore, disparities between these two words types may hinge on whether they involve overt bodily movements or not, as is the case with action verbs. For instance, De Grauwe and collaborators [47] observed activation in somatosensory and motor brain regions triggered when participants were exposed to a motion verb comprehension task in an FL (e.g., “to throw”). Consequently, it is plausible to speculate that the impact of gestures in FL vocabulary acquisition might be more pronounced with verbs than with nouns.

Considering the learning conditions, we anticipated (a) a positive effect (learning facilitation) when employing congruent gestures during FL word acquisition (congruent condition), and (b) a negative effect (learning interference) associated with gestures unrelated to word meanings (incongruent and meaningless conditions). This pattern of results may align with the postulates of the motor imagery account [122] and with previous studies that have illustrated both the benefits and drawbacks of using gestures in FL vocabulary acquisition [70,74]. 

Considering both translation directions, our general hypothesis was fitted to the principles of the Revised Hierarchical Model (RHM) [8]. The translation task allowed us to establish conclusions at the level of lexical and semantic access to the new learned words. First, we expected an asymmetrical effect, as seen in prior studies, with more efficient performance in backward translation compared to that in forward translation [8]. This prediction arises from the fact that forward translation necessitates more semantic processing than backward translation does, making it more challenging to translate from L1 to FL than the reverse. In general, it was predicted that in learning conditions where semantic processing is promoted, facilitation would be particularly noticeable in forward translation because it is semantically mediated [8,123,124]. In terms of potential interactions between the translation direction and the gesture conditions (congruent, incongruent, meaningless and no gestures), we anticipated that forward translation would be more affected by semantic congruence (congruent and incongruent conditions) compared to that in the condition with meaningless gestures.

### 2.6. Results of Experiment 1

Several factors treated as within-participant variables were entered in analyses of variance (ANOVA). These factors were the translation test (comprising forward translation and backward translation), learning session (comprising the first, second and third session), and learning condition (which included congruent, incongruent and meaningless gestures, and the no-gesture condition). The translation task order was neither significant nor did it interact with other factors, so this variable was not considered any further.

Response time analysis revealed that participants learned the new words properly across the three acquisition sessions. They responded faster in the last learning session compared to on the first day of training. The recall analysis showed a wider array of main effects and interactions. In line with the response time results, participants remembered more words at the end of the training compared to the first evaluation phase. Figure 3 illustrate the interaction between translation direction and learning condition in each learning session. In the forward translation direction (L1–FL), performance improved across sessions. The main effect of learning condition was modulated by the translation test (see Figure 4). The comparison between learning conditions showed better recall in the congruent gestures condition compared to the meaningless gestures condition. The meaningless, incongruent and no gesture conditions showed similar outcomes. The % Recall was lower in the meaningless condition compared to the condition without gestures. The backward translation direction revealed a similar session effect showing that the more participants trained, the more they learned. Comparing learning conditions, congruent gestures improved performance relative to that in the meaningless condition. There were no differences between the meaningless and incongruent conditions. Compared to the no-gesture condition, the recall was lower in the meaningless and incongruent gesture conditions.

When the learning of FL nouns was investigated across learning conditions, two primary effects emerged. When participants learned FL words in the congruent condition, there was a notably higher recall percentage compared to that in the meaningless condition. However, as shown in the backward translation task, the recall of FL words was lower in the incongruent and meaningless gesture conditions compared to that in the no-gesture condition. This pattern of results suggests that these conditions had a detrimental impact on the learning process. Altogether, these findings reveal two contrasting effects when comparing different methods of FL learning: facilitation and interference. Figure 4 shows the % Recall of nouns as a function of the translation direction and gesture conditions (learning session collapsed).

The motor imagery account might explain the facilitation effect observed with congruent iconic gestures [69]. The alignment of conceptual information between L1 words and gestures helped the acquisition of FL words. The facilitation effect could also be explained via the motor trace theory [65,66], as congruent gestures were familiar to participants which allowed the activation of motor traces associated with words. However, the interference effect observed in association with incongruent gestures cannot be explained from this account because, in this condition, gestures were as familiar to participants as in the congruent condition. Finally, as differences were evident between conditions involving gestures, the self-involvement explanation could not explain the observed pattern of results [59]. Notably, the magnitudes of learning interference in the meaningless and incongruent conditions reached a similar level. These findings could be indicative of the participants being involved in a dual-task scenario that heightened the complexity of information processing under these conditions.

### 2.7. Results of Experiment 2

The same analysis protocol and set of factors used in Experiment 1 were employed in Experiment 2. The sole distinction between the two studies was the category of words participants were tasked with learning. In Experiment 2, students acquired verbs instead of nouns. When the recall of nouns and verbs is considered, previous research has consistently shown that verbs tend to be more challenging to learn [106,110]. To address this inherent difficulty associated with verb acquisition, one potential approach is to incorporate gestures as a teaching strategy while learners are presented with new vocabulary. It has been suggested that verbs intrinsically contain a motor component in their semantic representation [105,107]. Consequently, in Experiment 2, we evaluated whether or not the use of gestures could alleviate the inherent difficulty associated with learning verbs vs. nouns.

When it comes to RTs, learners responded faster in the last training session compared to the first learning day. Regarding learning conditions, congruent gestures exhibited faster responses than those in the incongruent, meaningless and no-gesture conditions. No differences were found between any other learning condition. Recall was better in the last learning day compared to that at the beginning of training (see Figure 5 to observe the % Recall across sessions, translation directions and learning conditions). There was also a main effect of the translation direction that interacted with the learning condition factor. This effect is shown in Figure 6 (learning session collapsed). In the forward translation, the session effect previously described persisted. Better recall was observed in the congruent condition relative to that in the meaningless and no-gesture condition. No differences were found between the incongruent, meaningless and no-gesture conditions. The backward translation direction showed, again, the increase in percentages of recall across learning sessions. Better recall was observed in the congruent condition relative to the no-gesture and meaningless conditions. No differences were found between meaningless and incongruent gestures. However, compared to the no-gesture condition, the recall was lower in the incongruent condition and the meaningless condition.

The role of gestures in the acquisition of FL verbs was evaluated in Experiment 2. The observed pattern of results closely mirrored what we found in Experiment 1, which pertained to the learning of FL nouns.

Once again, we identified a facilitation effect stemming from the integration of gestures into the learning process. Specifically, congruent gestures proved to enhance the acquisition of FL verbs compared to that when learning without gestures or with meaningless gestures. However, as in Experiment 1, we also encountered an interference effect when participants were exposed to incongruent and meaningless gestures, as opposed to the no-gesture condition, which was particularly evident in the backward translation direction (FL–L1). The discussion section will provide a more in-depth explanation of these two effects, shedding further light on their underlying mechanisms.

### 2.8. Between-Experiment Comparison

To establish comparisons between word types, which constituted one of the primary objectives of this experimental series, we introduced the word type factor (nouns and verbs) in a new analysis along with the factors previously considered. In the forward translation direction, participants recalled more nouns than verbs. Also, the learning condition interacted with the word type factor. In the no-gesture and incongruent conditions, better recall was noted for nouns than verbs. However, no differences between nouns and verbs were obtained for congruent and meaningless gestures. Considering the backward translation, no main effects or interactions were significant. Figure 7 shows the results found for noun and verb recall in each of the learning conditions.

More detailed information about statistical data can be consulted in the study published by García-Gámez and Macizo in 2019 [111].

## 3. Review of Research from Our Lab: Experiment 3

In Experiment 3, the four learning conditions included in the previous experimental design (Experiments 1 and 2) were maintained but, in this case, two learning groups were evaluated. One of the groups learned new vocabulary through a physical engagement strategy by imitating the instructor’s gestures while the other group only observed the gestures produced by the instructor while learning. Participants included in Experiment 3 did not took part of Experiments 1 and 2 but had the same inclusion and exclusion criteria and proceeded from the same pool of students at the University of Granada. Concerning materials, only verbs were chosen as learning vocabulary, and the same action verbs used in Experiment 2 were used here. The main manipulations we introduced in this experiment focused on the experimental procedure. Consequently, specific predictions and hypotheses were derived from these manipulations. 

### 3.1. Procedure

The procedure employed in Experiment 3 closely resembled that used in Experiments 1 and 2. The differences focused on the division of participants between experimental groups and in the instructions participants were given during the vocabulary learning phase. 

There exists a debate surrounding the impact of self-generated movement compared to that of the mere observation of gestures in FL vocabulary acquisition. To directly address this issue, two experimental groups were included in Experiment 3. Participants were randomly divided into the “see” and “do” training groups. In the “see” learning group, learners were required to read aloud Spanish–Vimmi (L1–FL) word pairs while simultaneously observing and mentally envisioning themselves replicating the gestures depicted in a video. Only action verbs served as learning material in Experiment 3. In the “do” learning group, learners were also instructed to reproduce the words but in this experimental group, they were required also to physically mimic the gestures as they were presented on the screen (as in Experiments 1 and 2). For instance, in presence of the word pair “teclear-saluzafo” paired with a gesture congruent in meaning, they were expected to say this word pair aloud while concurrently generating a mental image of themselves moving fingers as if typing on a keyboard. This was maintained in the three gesture conditions (congruent, incongruent and meaningless conditions) but the participants did not perform any movement in the condition without gestures as occurred in the rest of the experiments and in the “do” learning group. The reason for instructing participants in the “see” learning group to mentally recreate the gesture was to ensure their engagement with the gesture and to prevent them from solely focusing on learning the words in Vimmi. While it was challenging to confirm a priori whether or not participants were indeed mentally visualizing the required gesture, post-experiment observations (i.e., the learning curve through training) indicate that participants did, in fact, follow the experimenter’s instructions and mentally executed the prescribed gestures.

At the end of each learning phase, participant learning was evaluated with a forward and backward translation task following the procedure described in Experiments 1 and 2. 

### 3.2. Predictions and Hypothesis

As previously mentioned, certain studies demonstrate enhanced learning outcomes associated with the active generation of gestures [10,97,103]. Conversely, in other research, no discernible difference is observed between the act of viewing gestures and the act of performing gestures by FL learners [24,55,102]. In this work, we controlled for several factors that might explain differences found in previous studies (type of learning material, type of gestures or word–gesture meaning relation). Unlike the vast majority of research in the field, we explore the effect of self-generated movements in an adult population [33,90,96,103]. It is crucial to emphasize this methodological distinction because adults generally possess more experience in executing actions compared to children. Furthermore, as adults gain extensive experience with the semantic content of words, it is expected that they possess richer semantic content and visual imagery associated with gestures [125]. Given this consideration, merely observing movements resembling action verbs might suffice for adult individuals to harness the beneficial effects of gestures in FL learning. Therefore, the act of observing gestures could potentially strengthen the connections between new vocabulary and the semantic system, much like actually performing the gestures [102]. In other words, we hypothesized that adult learners might not show any discernible difference between the act of viewing and the act of physically performing gestures in terms of FL vocabulary acquisition.

Regarding learning conditions, similar results to those of Experiment 1 and 2 are anticipated. In addition, if the mere act of observing gestures proved to be sufficient to enhance vocabulary acquisition, the outcomes would be consistent across both learning methods. Conversely, if active engagement in gestures during instruction, as seen in the “do” learning group, maximized learning, we would anticipate a higher learning rate in this group compared to that in the “see” learning group.

As in Experiment 1 and 2, when learning conditions (“do” vs. “see”) enhanced semantic processing, facilitation was be particularly noticeable in forward vs. backward translation because it was semantically mediated [8,123,124].

### 3.3. Results of Experiment 3

We conducted ANOVAs including the translation direction (forward and backward translation), learning session (first, second and third session), and learning condition (congruent, incongruent, meaningless and no gestures) that were treated as within-participants factors, while the learning group (“see”, “do”) was considered a between-groups variable in Experiment 3. Consistently with Experiments 1 and 2, the more participants trained, the faster and more accurate they were.

Regarding RTs, the main effect of the translation direction was significant, so participants were faster in backward than in forward translation. The learning condition was also significant and learners were faster in the congruent condition compared to the meaningless condition and the no-gesture condition. No differences were found between incongruent and meaningless gestures. Finally, marginal differences were obtained between meaningless gestures and the no-gesture condition with participants responding subtly slower to the meaningless condition. It might be highlighted that participants in the “do” learning group were faster than the ones who learned under the “see” instruction across learning sessions. 

Recall results showed that participants were more accurate in the congruent and no-gesture conditions compared to the meaningless condition. No differences appeared between the incongruent and meaningless conditions. Finally, compared to the no-gesture condition, better recall was observed in the congruent condition. In this case, the meaningless condition appeared to have the most detrimental impact on participants’ performance, whereas congruent gestures facilitated the learning process in comparison to that in the no-gesture condition. Differences emerged between learning groups with participants included in the “do” group being more accurate than participants in the “see” group. The learning group interacted with the learning condition and translation direction and hence, the analyses for each learning group were conducted separately. In the “see” group, congruent gestures produced a facilitation effect compared to meaningless gestures. Also, participants remembered more words in the no-gesture condition compared to the meaningless condition. Marginal differences were observed when the incongruent and meaningless conditions were compared being the meaningless gestures the most detrimental learning situation. The final comparison between congruent and no-gesture conditions showed a marginal effect. The “do” learning group revealed an interaction between the learning condition and translation direction. In this way, the effect of learning condition was explored for each translation direction separately. In forward translation, there was a facilitation effect associated with the use of congruent gestures. No differences were obtained between incongruent, meaningless and no-gesture conditions. The “see” group exhibited a learning interference effect in the presence of meaningless gestures that was not present in the case of the “do” group. Advantages were associated with congruent gestures compared to the no-gesture condition. Finally, in the backward translation direction, congruent gestures showed better recall patterns compared to meaningless gestures. Differences appeared between incongruent and meaningless gestures but similar results were obtained when the congruent and no-gesture conditions were compared (see Figure 8). Extended information about statistical data is provided in the paper published by García-Gámez and Macizo in 2021 [4].

## 4. Integrative Discussion and Theoretical Implications

It is commonly accepted that many cognitive processes are influenced by movement effects. In general, motion information has demonstrated a facilitative effect not only in learning contexts but also in other areas such as developmental disorders and aphasia treatments [126,127,128]. Particularly noteworthy are iconic gestures, referring to concrete entities or actions. Previous research has investigated the potential role of iconic gestures in memory consolidation, language comprehension and production [56,58], and in FL vocabulary acquisition [51,103]. The findings from the present series of studies contribute to enhancing our understanding of how gestures mediate the acquisition of vocabulary in an FL.

### 4.1. Gestures Produce Learning Facilitation and Interference Effects

Firstly, and highly importantly, regarding the impact of different gesture conditions in our studies, in the congruent gesture condition, where words and gestures shared semantic representations, there was a notable increase in word learning [48,74,76]. In general, participants exhibited more efficient patterns of recall with higher accuracy rates and faster response times when new words were acquired in presence of congruent gestures. This suggests that congruent gestures promote semantic processing, ultimately benefiting learning outcomes. Additionally, processing congruent gestures alongside word learning mitigated the negative effects of dual-task performance (processing words and gestures) [129,130]. This effect might support the motor-imagery theory concerning the effect of gestures in FL word learning [69]. The shared conceptual information between iconic gestures and L1 words appeared to enhance the learning process of FL words. In line with prior research, the facilitative effect of FL learning resulting from the processing of congruent gestures persisted consistently, irrespective of the learning group (either “see” training or “do” training). This suggests that simply being exposed to gestures is adequate to witness the beneficial effects of gestures on vocabulary acquisition in an FL [102]. While our behavioral studies did not provide evidence of brain activity, our results align with outcomes from various reports, which demonstrate that the mere observation of motion information triggers patterns of brain activity in the motor cortex akin to those observed during the actual performance of motor actions [98]. Consequently, processing gestures, whether through observation or performance, would have an impact on semantic/declarative memory via enriching the encoding of new words thanks to the incorporation of sensorimotor and procedural information into the memory representation of the new words [68]. Hence, congruent gestures can enhance the semantic processing of words and this pattern of findings align with the motor trace [65,66] and motor imagery accounts [69].

Results revealed that the incongruent and meaningless gesture conditions worsened performance compared to that in the no-gesture condition, revealing a learning interference effect (although no differences were found in the L1–FL translation of verbs in Experiment 2 and in the “do” learning group) [48,70,73,74,84]. The primary distinction between the incongruent and meaningless conditions when compared to the no-gesture condition was the level of engagement in a dual- versus a single-task learning context. This interference would be indicative of the difficulty associated with the integration of the meaning of words and gestures in working memory [84]. Hence, the learning and subsequent recall of FL words could be impeded by a conflict situation in working memory, stemming from the lack of alignment between the conceptual information conveyed by the gesture and the word’s meaning. The “see” learning group exhibited an additional interference effect (only in the L1–FL translation direction), with a poorer recall of words learned in association with meaningless gestures compared to the ones acquired in the no-gesture condition. In the case of meaningless gestures, as there was not a direct meaning associated with these movements, the interference could be a result of the activation of conflict motor traces associated with the gestures and the action verbs. Therefore, the use of gestures may have a negative impact in the learning process, particularly when meaningless gestures are performed by instructors and no repeated by learners. In addition, results revealed no differences between these two conditions (incongruent and meaningless conditions) (except a small effect found in the “see” learning group included in Experiment 3). Specifically, participants found themselves in a dual-task situation where they were required to simultaneously process both the gesture and the corresponding L1 word while learning the FL word. This dual coding demand resulted in a cost, aligning with findings from other studies that underscore the challenge of encoding the meaning of an FL message when learners are concurrently involved in another task [129,130,131]. 

Coming back to the theories explaining the potential role that gestures have on FL vocabulary acquisition, the self-involvement account fails to explain the global pattern of results. In line with this perspective, gestures can be seen as a means to enhance the participant’s engagement in the learning task. Consequently, whenever gestures are integrated into the learning process, one would expect to observe an improvement in word acquisition. However, in our experiments, clear differences were observed among gesture conditions; also see [59]. Considering the incongruent and meaningless gestures, both the motor trace [65,66] and motor imagery accounts [69] establish specific predictions about the learning results. The motor trace theory supports that only when learners perform familiar gestures, a facilitation effect could be found. Hence, this theory could suggest reduced performance in the meaningless and no-gesture conditions compared to that in the congruent and incongruent conditions. On the other hand, the motor imagery account might entail higher learning interference associated with incongruent gestures followed also by reduced performance in the meaningless conditions compared to the no-gesture condition. The discrepancy between the meaning conveyed by the gestures and the meaning of the FL words being learned is more pronounced in the incongruent condition, leading to this distinct effect. The findings found in the incongruent and meaningless conditions of our experiments might not totally align with any of the theoretical predictions described above, although the motor imagery theory could accommodate to a greater extent our pattern of results.

In summary, congruent iconic gestures proved to be a powerful tool to implement in the learning process of new vocabulary. This effect was observed when learners produced the gestures through imitation and also when they merely observed the instructor’s gestures. On the other hand, incongruent and meaningless gestures produced interference in learning, which could be explained as a consequence of the dual-task situation in which participants were immersed. These facilitation and interference effects could be accommodated within the motor imagery account although the additional difficulty related to the presentation of incongruent gestures compared to meaningless gestures did not appear in our studies.

### 4.2. Gestures in Noun and Verb Learning

Our studies revealed differences between the learning of nouns and verbs. Typically, nouns exhibited a higher learning rate than verbs did, primarily due to the greater semantic content of nouns [110,132,133]. In our series of studies, the results consistently indicated the same trend. In the absence of gestures (the no-gesture condition), better recall was registered for noun than verb learning. When gesture impact was analyzed considering word type (nouns and verbs), similar interference and facilitation effects were found. Importantly, the data revealed an interesting pattern: the consistent additional difficulty associated with verbs learning diminished when congruent gestures were incorporated into the learning protocol. In essence, gestures used in the process of acquisition of vocabulary in an FL appeared to alleviate the intrinsic challenges typically associated with learning verbs. Once more, as the noun–verb distinctions vanished when gestures conveyed the same meaning as the FL words being acquired, this discovery aligns with the motor imagery perspective. Interestingly, there is another hypothesis about the role of gestures on FL learning that can explain the positive effect associated with learning verbs accompanied by gestures. In fact, the gestures for the conceptualization hypothesis [134] points out that gestures have their origins in actions that encompass bodily movements and motor-related content. Importantly, the meaning of verbs inherently encompasses motor-related information. Therefore, gestures directly participate in simulating the meaning of verbs, consequently enhancing the acquisition of this particular category of words. This hypothesis aligns well with the effects observed in our study, highlighting the unique role of gestures in verb learning.

### 4.3. The Role of Gestures in the Establishment of Lexical–Semantic Connections

The translation task participants performed at the end of each learning session gave us information about the development of lexical and semantic links through the acquisition process. To better understand these effects, our results can be accommodated within the RHM hypothesis [8]. In short, this model proposes the existence of a network of interconnected elements, including L1 and FL words, and a shared semantic system. However, the strength of the links between these elements is modulated by the learning stage of FL vocabulary. In initial moments, the semantic system–FL word connections are relatively weak, leading FL learners to predominantly rely on a lexical connection when processing from FL to L1. As learners become more proficient in in the FL, the lexical–semantic connections increase, while the importance of lexical links diminishes. This model has garnered substantial support from previous research. For instance, the model finds support in studies implementing translation tasks where asymmetric performance was found in unbalanced bilingual speakers. These individuals tend to exhibit faster performance in backward translation (which involves the lexical route from FL to L1) compared to forward translation (which relies on the semantic route from L1 to FL). This phenomenon has been documented in research by Kroll and Stewart [8] and contributes to our understanding of how bilingual speakers process and translate words between two languages depending on their FL proficiency levels. In our studies, the learning procedure including congruent gestures proved to foster the semantic route of processing. Previous research has demonstrated that the use of gestures as a learning strategy enhances the encoding of new words by integrating sensorimotor networks and procedural memory into the semantic/declarative memory linked with word meanings [68]. Consequently, gestures play a role in enriching the semantic processing of FL words. The findings from our current studies provide evidence suggesting that the use of gestures is associated with semantic connections in FL learning. As previously mentioned, gestures effectively relieve the difficulty typically linked to the learning of verbs vs. nouns. This effect was particularly noticeable in forward translation, a task that relies more heavily on semantic mediation compared to backward translation (as discussed in [135]). Furthermore, our exploration of the linguistic properties of nouns and verbs in our material revealed an important difference in concreteness level, with nouns reflecting higher levels of concreteness than verbs. Concreteness is a variable known to influence FL vocabulary learning, with concrete words activating the semantic system more strongly than abstract words do. This difference in activation makes concrete words more readily acquired by FL learners [136].

### 4.4. Gesture Observation and Imitation

In our third experiment, our primary focus was to directly compare the effects of gesture performance and observation when acquiring vocabulary in an FL. Participants were randomly included in two groups. In the “do” learning group, participants themselves produced the gestures associated with the FL words (as in Experiments 1 and 2), but in the “see” group, learners solely observed the gestures produced by the instructor. As experimenters, we took the risk of giving participants in the “see” learning group the instruction to imagine themselves performing the gestures. We carried this out to ensure that participants in the “see” condition did not exclusively focus on learning FL words while neglecting the processing of gestures. The results unveiled a higher recall of FL words in the “do” learning group compared to that in the “see” learning group (a 12% advantage). Further, participants in the “do” learning group retrieved FL words significantly faster than those in the “see” learning group did. The approach of emphasizing self-generated gestures during training contributed to an enhancement in the retrieval of new vocabulary in an FL. It is possible that instructing participants in the “see” group to visualize the gestures may have mitigated the potential for a more uniform learning group effect. However, previous studies comparing actual physical manipulation with an imagined manipulation condition have consistently shown superior outcomes when participants actively engage in motor activity [33].

Concerning verb processing, Hauk and colleagues [107] and Buccino and colleagues [137] noted that when participants were directed to read action verbs or simply required to observe motor actions, motor cortex areas associated with body regions involved in actual movement were activated in accordance with a somatotopic organization. In our study, the results obtained when participants were instructed to only observe the gestures on the screen might indicate that incongruent and meaningless gestures could produce an interference in learning due to the mismatch between the gestures and words semantics [138]. This interference was partially reduced when participants were instructed to engage in the physical activity, as in the “do” learning group. In this case, less interference was observed in the forward translation direction but it persisted in the backward translation task. At this point, we may wonder about the cognitive mechanisms behind this reduced interference effect when learners produced the required movements. In previous studies, advantages were found in association with real movement production in problem solving [139]. The beneficial effects were attributed to a reduction in working memory load associated with actual movement performance. Continuing this line of thought, Experiment 3 might reveal that executing gestures during the learning process could attenuate the cognitive effort demands on working memory. Consequently, this could alleviate challenges in integrating semantically incoherent information arising from the words’ and gestures’ meanings in the meaningless and incongruent gesture conditions. In fact, the availability of working memory resources has been pointed as a crucial factor when participants are required to solve conflict situations in an L1 and L2 [140].

As previously mentioned, in our studies, participants recalled more words and they responded faster in the FL–L1 translation task compared to the L1–FL translation direction in the no-gesture condition. This outcome underscores the greater difficulty associated with forward translation compared to backward translation, even when gestures are not used during the learning of FL words. Furthermore, concerning the “do” learning strategy, learning interference effects were attenuated in the L1–FL translation direction compared to those observed in the LF–L1 direction (see Figure 8). Specifically, the interference observed in the meaningless condition in the FL–L1 translation vanished in the L1–FL direction, showing similar execution when words were learned with meaningless or incongruent gestures, or in isolation. Again, the key distinction between the translation direction tasks lies in their difficulty levels. As previously noted, the forward translation task is more challenging for learners, especially in the initial stages of vocabulary acquisition. This could be explained by the increased cognitive demands involved in semantic vs. lexical processing [8]. In this sense, we could argue that the impact of gestures on vocabulary acquisition might be in line with task difficulty. For instance, Marstaller and Burianová [36] observed that participants obtaining lower scores in working memory capacity showed greater benefits associated with the use of gestures in a letter memorization task. Hence, gesture use facilitated letter recall in those participants for whom the task was more challenging. In our study, actual physical engagement in the task mitigated the learning interference produced due to the word–gesture mismatch, particularly when task demands were higher, which was the case in the forward translation task.

At this juncture, one might wonder why the reduced interference effect found in the forward translation task in the “do” learning group did not appear in the “see” group. The learning condition did not interact significantly with the translation direction in this group of participants. In addition, there were no differences in performance as a function of the translation direction. One could argue that this effect is, in fact, an advantage because participants did not exhibit the typical difficulty found when the forward and backward tasks are confronted [8]. However, although we acknowledge that we do not possess a definitive explanation for this phenomenon, tentative explanations prompt us to interpret it as a consequence of the way verbal material was presented in our study, specifically the presentation of a Spanish word followed by its Vimmi translation, L1–FL word pairs. This might have enhanced a lexical coding strategy between languages regardless of the required translation direction. In contrast, the physical production of gestures could have enhanced the semantic processing of the new words in the “do” learning group [69,82]. This promotion of conceptual processing might be behind the more pronounced translation direction effects in participants included in this learning strategy [8,125].

In general, physical engagement in the learning task led to recall facilitation compared to the observation of movements. When congruent gestures were used as learning material, advantages were obtained in both learning groups. In addition, the learning interference associated with incongruent and meaningless gestures decreased when learners self-performed the gestures. This might indicate that while congruent gestures could produce beneficial effects even through observation, incongruent and meaningless gestures are especially sensitive to interference specially when students do not have to imitate the gestures. 

### 4.5. Final Research Questions

On another note, the pattern of results found in Experiment 2 can be compared with the learning outcomes of the “do” learning group in Experiment 3, as both experiments employed the same design and both used verbs as learning material. In general, both studies revealed similar main effects and interactions between variables. However, some differences were noted in the final outcomes. Specifically, in Experiment 2, in the backward translation direction, when the congruent and the no-gesture conditions were compared, a learning advantage effect was obtained thanks to congruent gesture performance in the backward translation task. In contrast, participants following the same procedure in Experiment 3, that is, the “do” learning group, did not show differences between the congruent and no-gesture conditions in this translation direction. We speculate that a potential explanation for these discrepancies could be the variations in the overall rate of word recall across the studies. Generally, participants’ performance was lower in Experiment 2, characterized by a poorer recall of words and delayed response times, in contrast to that of participants in Experiment 3. Consequently, participants in Experiment 2 might have had greater potential to benefit from engaging in congruent gestures during FL word learning.

The findings from this series of studies have direct applications in the everyday language learning practices of both learners and instructors. Language instructors can significantly benefit from staying updated on novel research on effective learning strategies that enhance the language acquisition process. Specifically, congruent iconic gestures have proven to be an effective tool, particularly in the acquisition of verbs in an FL. Given that verbs are generally more challenging to acquire than nouns, utilizing this strategy has the potential to significantly facilitate the learning process. The mere observation of congruent gestures has demonstrated notable efficacy, suggesting that language instructors could adopt a teaching strategy centered around self-performed gestures congruent in meaning with the new words. This approach could be further reinforced by encouraging students to mimic the instructor’s gestures. Engaging learners in this imitation process has shown positive effects, notably in reducing working memory load and alleviating the adverse impacts of gesture–word incongruence. As new technologies increasingly accompany us over time, it is crucial to note that these findings have implications beyond traditional classroom settings. These implications extend not only to scenarios where instructors can perform movements themselves or through virtual agents but also in online or virtual classes, which have emerged to enhance accessibility to education, especially in the last few years.

This series of studies also presents various limitations that could be explored and improved upon in future research. First, the sociodemographic context of learners can significantly influence the effect of gestures on FL acquisition. Notably, the majority of research, including present and previous studies, has predominantly focused on evaluating children or young-adult populations, while relatively less attention has been given to older learners. Moreover, participants in our studies possessed a high educational level. Future research endeavors should aim to bridge this gap by targeting diverse age groups and individuals with varying educational backgrounds. In this way, data on the role of gestures on vocabulary learning would be more generalizable. Additionally, contextual factors such as cultural influences and the physical environment where instructions are developed, whether in a controlled laboratory setting or in regular classroom settings, can impact performance. However, we know that gestures have also proven their effectiveness in more natural instructional settings (e.g., [52,76]). Other intrinsic characteristics or the psychological state of individuals may also contribute as influential factors shaping behavioral outcomes. As mentioned in the methods section, we controlled for participants’ proficiency in any second language. It is crucial to note that previous research suggests that learning a L3 might be easier for bilingual participants than learning a L2 (see [117] for a review). This underscores the potential modulation of gestures’ effects by speakers’ language proficiency. Although we did not incorporate additional measures on cognitive-related factors, such as working memory [84], these measures might significantly influence language learning. Including additional control tasks could have provided further insights into this aspect. Finally, regarding the experimental design, a significant factor to expand the potential generalization of our findings could be the introduction of a long-term evaluation task to assess the evolution of word recall in the different learning conditions over time.

Focusing on our data, future research endeavors could delve deeper into the relationship between incongruent and meaningless gestures. Both conditions appear to demonstrate a similar pattern of processing, irrespective of whether they represent a familiar or unfamiliar gesture. This would be against the motor trace and motor imagery predictions. Additionally, designing a long-term assessment program would be effective to determine how the recall of new words evolve over time when learners acquire new vocabulary across conditions. Moreover, the effect of gesture imitation and gesture observation can be tested in other lexical categories such as adjectives. Finally, approaches rooted in conceptual material processing, such as using images alongside words, as seen in Tellier [103], have also shown promise in supporting FL learning. A future direction for exploration could involve assessing the potential cumulative benefits of combining these two instructional aids (gestures and images) on FL vocabulary acquisition. An additional crucial factor to consider in future studies is the presence of individual differences among learners, which can influence the effectiveness of specific strategies for FL acquisition [141,142,143]. Furthermore, previous studies have delved into brain activation and activity modulation resulting from the use of gestures in the FL vocabulary learning process [36,43,70,144]. Further research in this field is imperative to unravel the underlying processes responsible for both the advantageous and detrimental effects associated with gesture–word congruency or incongruency. Future investigations should aim to incorporate a diverse range of neuroimaging techniques and analyses, such as electroencephalography (EEG) featuring event-related potentials (ERPs) and time frequency analysis, as well as functional magnetic resonance imaging (fMRI). These approaches could significantly contribute to the elucidation of how gestures facilitate the formation of new memory traces, consequently aiding in the acquisition and retrieval of new words when gestures are introduced as a learning tool.

## 5. Conclusions

As mentioned in Section 1, the theories explaining the potential role gestures have on FL vocabulary learning are not mutually exclusive. In fact, these accounts highlight different aspects of gestures’ effect on FL learning. Accompanying a word with a gesture may enhance self-involvement (thereby increasing attention to FL learning), establish a meaningful motor trace and/or evoke a semantic visual image that is integrated with the word’s meaning. However, considering our experimental results, the motor imagery account seems to be more aligned with our pattern of outcomes considering the different learning conditions. As the theory predicts, we found that only the use of gestures congruent in meaning with the target words facilitates vocabulary acquisition. This advantage extends to the realm of verb learning, where congruent gestures alleviate the difficulties encountered compared to noun acquisition. In addition, this effect is more pronounced when learners actively engage in gesture production. Moreover, gesture production seems to counteract potential adverse effects linked to unrelated gestures. Considering meaningless and incongruent gestures, the motor imagery perspective predicts greater interference to be associated with incongruent gestures. However, this prediction was not reflected in our experimental results, and further investigation is necessary to fully elucidate how unrelated meaningful and meaningless gestures interact with vocabulary acquisition. Taking into account the information obtained in this series of studies, we would like to highlight that when deciding between various FL learning methodologies incorporating gestures, our recommendation is to implement a training protocol that encourages participants to generate gestures aligned with the words they are learning. Finally, we would like to point out that learning a new language is not as easy as rubbing a magic lamp but can be easier with the appropriate learning methods.

## Figures and Tables

**Figure 3 brainsci-13-01712-f003:**
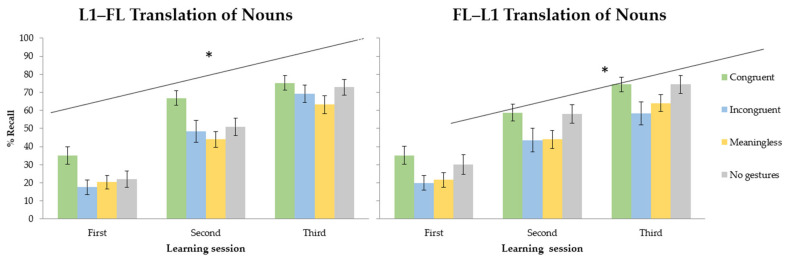
Recall percentages (% Recall) registered in Experiment 1 during forward and backward translation task of nouns considering learning sessions (first, second and third) and learning conditions (congruent, incongruent, meaningless and no gestures). Vertical lines represent standard errors. * *p* < 0.05.

**Figure 4 brainsci-13-01712-f004:**
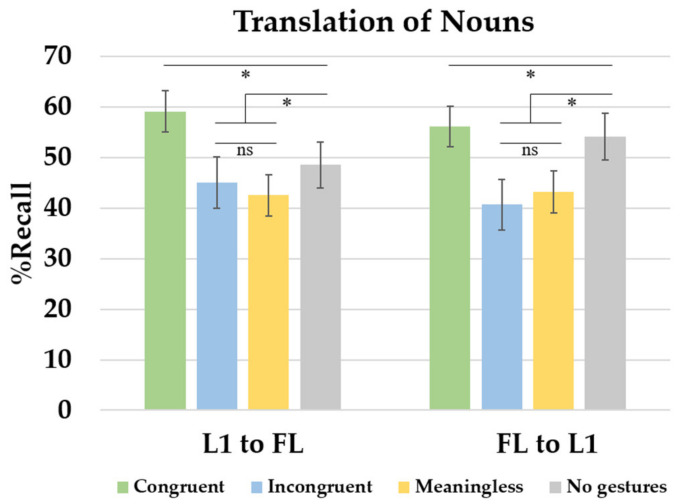
Recall percentage (% Recall) of nouns in each translation direction (L1 to FL; FL to L1) and learning condition (congruent, incongruent, meaningless and no gestures). Vertical lines represent standard errors. * *p* < 0.05, *^ns^* *p* > 0.05.

**Figure 5 brainsci-13-01712-f005:**
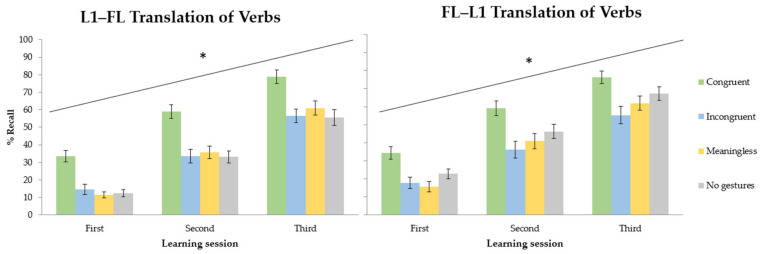
Recall percentages (% Recall) obtained in Experiment 2 during forward and backward translation task of verbs considering learning sessions (first, second and third) and learning conditions (congruent, incongruent, meaningless and no gestures). Vertical lines represent standard errors. * *p* < 0.05.

**Figure 6 brainsci-13-01712-f006:**
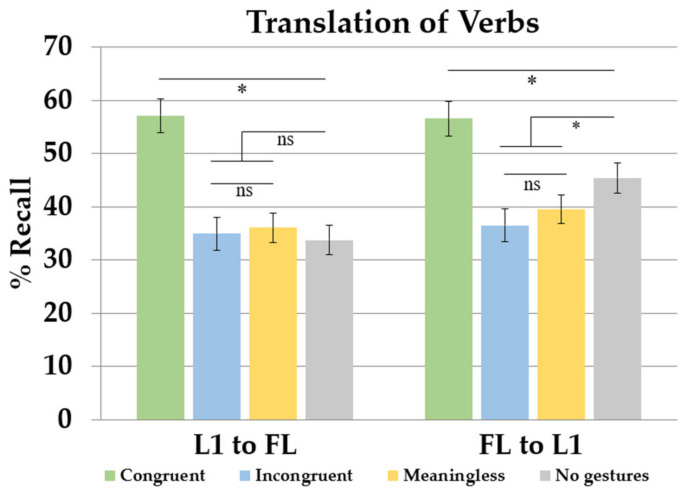
Recall percentage (% Recall) of verbs in each translation direction (L1 to FL; FL to L1) and learning condition (congruent, incongruent, meaningless and no gestures). Vertical lines represent standard errors. * *p* < 0.05, *^ns^* *p* > 0.05.

**Figure 7 brainsci-13-01712-f007:**
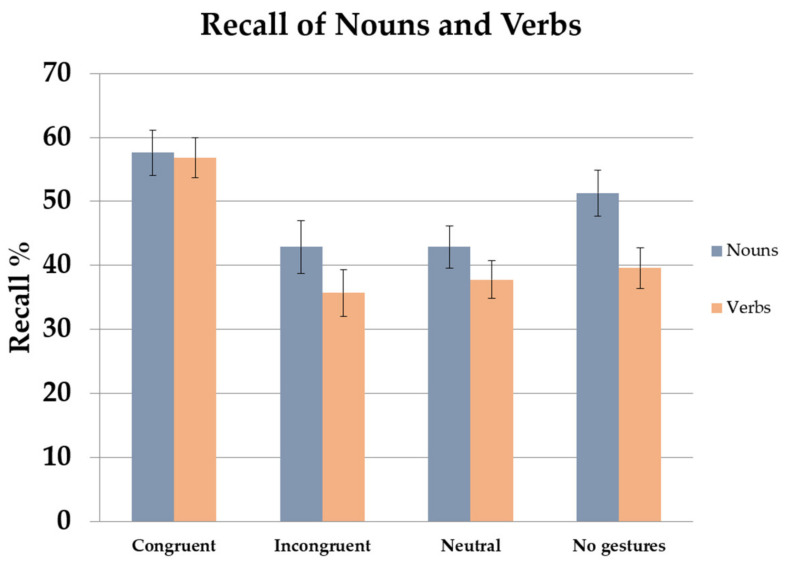
Recall percentages (% Recall) for nouns (Experiment 1) and verbs (Experiment 2) presented across learning conditions (congruent, incongruent, meaningless and no gestures). Vertical lines represent standard errors.

**Figure 8 brainsci-13-01712-f008:**
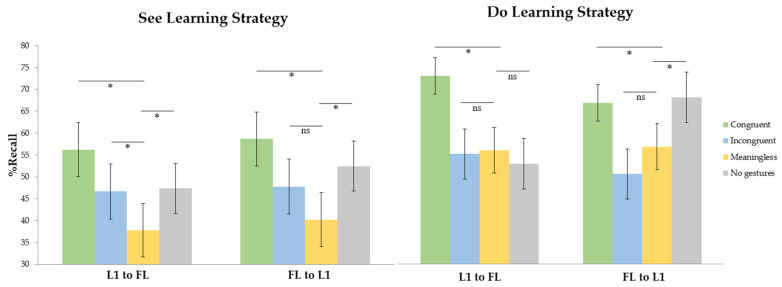
Recall percentage (% Recall) of the ‘see’ (left graph) and ‘do’ (right graph) learning groups in each translation direction (L1 to FL; FL to L1) and learning condition (congruent, incongruent, meaningless and no gestures). The standard error is plotted in vertical lines. * *p* < 0.05, *^ns^* *p* > 0.05.

## Data Availability

Publicly available datasets were analyzed in this study. This data can be found here: https://doi.org/10.17605/OSF.IO/TSGX4.

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
