# Peer review of "Gestures as Scaffolding to Learn Vocabulary in a Foreign Language"

_brainsci, 2023, doi:10.3390/brainsci13121712_

Round 1

Reviewer 1 Report

Comments and Suggestions for Authors

This is a review of brainsci-2716999, “Gestures as scaffolding to learn vocabulary in a foreign language.” The authors report the results of three experiments investigating whether gesturing (seeing or producing) during learning of foreign language vocabulary helps with learning. I found the manuscript to be very strong in terms of its theoretical motivation—the authors do a nice job laying out the alternative points of view about why gestures might help, and how those different views make different predictions about when they should help the most. I would like to see the authors’ return to the theoretical positions in the conclusions section of the paper and offer some concluding thoughts about which of the models seems best supported by their data.

My main suggestions to improve the paper have to do with the presentation. I come from a social science (rather than biological science field) and there may be different norms across fields, but in general I really did not care for the way this paper combined all three experiments together rather than describing each in turn. My preference would be to give a very short overview of the three studies at the end of the intro, and then include the detailed description of hypotheses along with the method and results of that experiment, in separate sections for each experiment. This would help the narrative of the paper build, with the reader clearly seeing why each experiment was done to parse out and further clarify the results of the previous experiment (which have already been given). As is, when you get to “experiment 2” or “experiment 3”, it is difficult to remember what the purpose of that experiment was and then in the results, it is difficult to remember what the method of that particular experiment was. I just found it very difficult to follow and read this paper with the presentation of the three experiments all mixed up like this, and if the journal allows a presentation that is more organized around each experiment in turn, that would be highly preferable in my opinion.

I also found the paper to be too long, and saw many sentences and ideas that could be cut. Much of the opening paragraph, for example, could be cut. It is not necessary to say that “research is necessary to investigate this problem” as that is self-evident. Simply state what the problem is—e.g., Adult learners often want or need to learn a foreign language, but what is the best way to do so?—and then launch into the literature review.   As another example, the first paragraph of the section “movements and languages” seems unnecessary. The definition of representational gestures (which are the only ones that are really necessary to define given their interest in the study) can be incorporated into the following paragraph. I do not think it is necessary to review all types of gestures, and the authors should instead focus their definitions to only those typologies that are relevant to the present study.

The authors should include more numerical evidence for their statistics—it would be difficult to include these findings in a meta-analysis without means and standard deviations (at a minimum) provided for each condition. While the figures are nice, it is difficult to judge effect sizes without exact statistics for each analysis.

A few methodological details seemed lacking (or maybe I just didn’t see them in the manuscript—as stated above, I had a difficult time following the method and knowing where to look for specific details and pieces of information regarding each experiment).

·      Participants were told they were video-recorded to make sure they followed the instructions—were they actually? What does it mean that “post-experiment observations indicatet hat participants were following the instructions” (p. 16, line 772)? Did the authors check to see if the instructions were correctly followed? 

·      Within a block, were the conditions blocked or inter-mixed? That is were the first ten pairs presented in condition A, the next 10 in condition B, and so on? Or were they all mixed up across the 40 trials?

·      How far apart were the learning sessions? From reading the description of the vocabulary learning phase, I didn’t understand that there were three sessions. Is the evaluation phase happening each time after the learning phase? Was the order of the forward vs. backward translation tasks kept the same in each evaluation session for a particular participant? Is Figure 4 (for example) averaging across the three learning sessions?

The discussion would benefit from some subsections or at least more focused topic sentences that clearly convey the takehome messages and findings. There are a lot of results here across the three experiments, and I found it difficult to really say what the main findings are after reading the paper. The Dicsussion starts okay by saying there are “several pieces of evidence to support the conclusion {that iconic gestures enhance semantic processing), but then the paragraphs that follow do not clearly lay out those pieces of evidence. Instead, they seem to go back to the literature and what has been shown there. The discussion should be more squarely focused on describing the findings of this study first, and then using the previous literature to contextualize that finding, pointing similarities or differences. At present, the discussion does not really summarize the main findings of this set of experiments and so does not leave a clear sense of what has been discovered here!

Author Response

Reviewer: 1

This is a review of brainsci-2716999, “Gestures as scaffolding to learn vocabulary in a foreign language.” The authors report the results of three experiments investigating whether gesturing (seeing or producing) during learning of foreign language vocabulary helps with learning. I found the manuscript to be very strong in terms of its theoretical motivation—the authors do a nice job laying out the alternative points of view about why gestures might help, and how those different views make different predictions about when they should help the most.

  1. I would like to see the authors’ return to the theoretical positions in the conclusions section of the paper and offer some concluding thoughts about which of the models seems best supported by their data.

We thank the reviewer for the positive feedback on the theoretical impact of our paper. In addition, we thank the reviewer for this first comment, guided to further improve the theoretical quality of our manuscript. In the conclusion section (please, see revised manuscript, pp. 25-26), we have developed the idea that the theories outlined across the paper are not mutually exclusive, and it is plausible that different aspects related to gesture use can work together to ultimately enhance the learning process. Additionally, in the discussion of this manuscript, each theory is analyzed point by point to examine how they align or not with the observed pattern of data across studies (please, see pp. 20-25). Moreover, following the reviewer's suggestion, we have detailed how the motor imagery account explains our findings to a greater extent than other theoretical approaches. Finally, we have been fair in pointing out the weaknesses of this theory in explaining some of our outcomes.

  1. My main suggestions to improve the paper have to do with the presentation. I come from a social science (rather than biological science field) and there may be different norms across fields, but in general I really did not care for the way this paper combined all three experiments together rather than describing each in turn. My preference would be to give a very short overview of the three studies at the end of the intro, and then include the detailed description of hypotheses along with the method and results of that experiment, in separate sections for each experiment. This would help the narrative of the paper build, with the reader clearly seeing why each experiment was done to parse out and further clarify the results of the previous experiment (which have already been given). As is, when you get to “experiment 2” or “experiment 3”, it is difficult to remember what the purpose of that experiment was and then in the results, it is difficult to remember what the method of that particular experiment was. I just found it very difficult to follow and read this paper with the presentation of the three experiments all mixed up like this, and if the journal allows a presentation that is more organized around each experiment in turn, that would be highly preferable in my opinion.

Thank you for the comment aimed at improving the accessibility of the manuscript to researchers from different scientific fields. In our manuscript, we provided a combined description of the studies because of the methodological similarities between them. We thought that this would make it easier to highlight the differences between experiments. Consequently, following the presentation of factors that remained consistent across experiments in the “current series of studies” section (please, see p. 8), we provided detailed explanations of the critical manipulation of each experiment used to evaluate the question at hand.

However, we agree with the reviewer. The way in which the information is presented may be confusing for a reader unfamiliar with the methodology used in the manuscript. Thus, following the reviewer recommendation, we have separated the information concerning each of the experimental studies:

Firstly, the contents initially placed in the “current study section” at the end of the introduction has been relocated. Instead, we have added a brief description of the series of studies (please, see p. 8). Experiments 1 and 2 are presented together because the only distinction between them is the type of word used as learning material (nouns or verbs). We have emphasized that this is the unique difference between Experiments 1 and 2. Subsequently, we address participant characteristics, experimental conditions, materials, procedures, predictions, hypotheses, and results for both Experiments 1 and 2 (please, see pp. 9-17). Following this, in the 'Experiment 3' section, we detail the differences between this study and those reported previously (please, see pp. 17-20). New manipulations are defined, along with the predictions, hypotheses, and results of the third study. Again, we thank the reviewer for this comment aimed at making the manuscript easier to read for scientists in different fields of research.

  1. I also found the paper to be too long, and saw many sentences and ideas that could be cut. Much of the opening paragraph, for example, could be cut. It is not necessary to say that “research is necessary to investigate this problem” as that is self-evident. Simply state what the problem is—e.g., Adult learners often want or need to learn a foreign language, but what is the best way to do so?—and then launch into the literature review.   As another example, the first paragraph of the section “movements and languages” seems unnecessary. The definition of representational gestures (which are the only ones that are really necessary to define given their interest in the study) can be incorporated into the following paragraph. I do not think it is necessary to review all types of gestures, and the authors should instead focus their definitions to only those typologies that are relevant to the present study.

We would like to thank to the reviewer for this comment. In response to this suggestion, we have revised both the opening paragraphs of each section of the manuscript; we have also carefully reviewed the 'movements and language' section (please, see pp. 2-3). In addition, a thorough revision of the entire manuscript has been carried out. We have removed redundant information and, consequently, the length of the manuscript has been substantially reduced.

  1. The authors should include more numerical evidence for their statistics—it would be difficult to include these findings in a meta-analysis without means and standard deviations (at a minimum) provided for each condition. While the figures are nice, it is difficult to judge effect sizes without exact statistics for each analysis.

Thank you for your comment. The aim of this manuscript was to present an integration of previous work done in our lab on the role of gestures in foreign language vocabulary learning. The individual studies, which have already been published, did not allow for an integrated theoretical elaboration to establish a global discussion on the topic. This was done in the current manuscript.    

Since the main contribution of this study compared to previously published individual experiments was theoretical, we decided not to include specific statistics. On the contrary, we thought that the graphical description of the results would be adequate for the understanding of the theoretical ideas conveyed in the manuscript (the graphical representation of previous results has not been previously published as they have been designed specifically for this paper).

The reviewer is completely correct in claiming that the non-inclusion of specific statistics in this article would preclude the performance of additional studies (e.g., meta-analysis). In order to solve this issue without blurring the theoretical goal of the manuscript and to avoid duplicity in reporting statistical data already published, in each results section of the current manuscript, we explicitly and carefully indicate the specific source where to go to obtain exact statistical values of the analyses conducted in each experiment (please, see p. 17, line 766-767 for Experiments 1 and 2, and p. 20, lines 882-884).

  1. A few methodological details seemed lacking (or maybe I just didn’t see them in the manuscript—as stated above, I had a difficult time following the method and knowing where to look for specific details and pieces of information regarding each experiment).

Thank you for the comment, in response to other reviewer remarks the explanation of the methodology used in our line of research has been clarified throughout the revised version of the manuscript.

  1. Participants were told they were video-recorded to make sure they followed the instructions—were they actually? What does it mean that “post-experiment observations indicatet hat participants were following the instructions” (p. 16, line 772)? Did the authors check to see if the instructions were correctly followed? 

Despite informing participants that sessions may or may not be recorded and having a camera installed as part of the experimental settings, no images were actually recorded. The final results pattern, showing differences between experimental conditions, especially the distinctions between the “do” and “see” groups, can confirm the active participation of the participants in the task. This information has been clarified in the revised version of the manuscript (p. 11).

  1. Within a block, were the conditions blocked or inter-mixed? That is were the first ten pairs presented in condition A, the next 10 in condition B, and so on? Or were they all mixed up across the 40 trials?

Thank you for this comment. We have rewritten part of the paragraph explaining the experimental procedure to clarify how the learning material was presented. The gesture conditions presentation was blocked. This information has been added to the revised version of the manuscript: “The learning phase lasted approximately 1 hour per session. We employed a stimulus presentation procedure organized by experimental conditions (blocked stimuli presentation per learning condition) [90]. This blocked design was adopted to minimize the cognitive effort associated with constantly switching between conditions where participants had to perform gestures and those without gestures. A single learning block contained the 40 L1-FL pairs (10 from each learning condition). Each word was presented 12 times resulting in each participant receiving 12 blocks and hence, 480 trials.” (p. 11).

  1. How far apart were the learning sessions? From reading the description of the vocabulary learning phase, I didn’t understand that there were three sessions. Is the evaluation phase happening each time after the learning phase? Was the order of the forward vs. backward translation tasks kept the same in each evaluation session for a particular participant? Is Figure 4 (for example) averaging across the three learning sessions?

Thank you for this comment. We have clarified this information in the procedure section of Experiments 1 and 2: “Participants were exposed to 3 learning and concurrent evaluation sessions that took place on three consecutive days with a delay of 24 hours between sessions.” (p. 11).

Regarding translation tasks, we have reviewed the information previously given in the manuscript to clarify this question: “To prevent any potential order/practice effects, the order of presenting the translation tests was randomized across the three training sessions and among participants. In each translation task, participants received 40 Spanish words and the 40 Vimmi words for forward and backward translations respectively.” (p. 11).

            Finally, in p. 14 lines 661-662, it is indicated that Figure 4 reflects the interaction between learning condition and translation test.

We thank the reviewer for this comment and the previous one with the aim of improving the understanding of the learning methodology used in our work.

The discussion would benefit from some subsections or at least more focused topic sentences that clearly convey the takehome messages and findings. There are a lot of results here across the three experiments, and I found it difficult to really say what the main findings are after reading the paper. The Dicsussion starts okay by saying there are “several pieces of evidence to support the conclusion {that iconic gestures enhance semantic processing), but then the paragraphs that follow do not clearly lay out those pieces of evidence. Instead, they seem to go back to the literature and what has been shown there. The discussion should be more squarely focused on describing the findings of this study first, and then using the previous literature to contextualize that finding, pointing similarities or differences. At present, the discussion does not really summarize the main findings of this set of experiments and so does not leave a clear sense of what has been discovered here!

We would to thank to reviewer 1 for this comment. Following our intention of mixing the experimental results to present a global message instead of separated experiments, we did a similar distribution of the discussion (interrelation of theoretical conclusions across the three experiments). However, as the reviewer indicates, it is possible that the information is more clearly presented by separating it in distinct subsections. Following the recommendation of the reviewer, the discussion section has undergone a complete restructuring, with the information divided into subsections addressing the main issues explored in this series of studies. Moreover, duplications and redundant information have been removed (see pp. 20-25).

At the end of each section, clear and direct messages are provided, offering readers information about our findings and their interpretation. As recommended, the conclusion has also been reviewed, incorporating the reviewer's comments and suggestions. (see pp. 25-26).

We would like to explicitly thank reviewer 1 for the careful analysis of our manuscript and for all the helpful comments and suggestions aimed at improving the quality of our manuscript.

Reviewer 2 Report

Comments and Suggestions for Authors

Dear authors, thank you for conducting such a comprehensive study. I have the following concerns:

Firstly, in the Abstract, there is too much detailed information about this study. It is suggested that the Abstract be concise and focus on the rationale, methodology, results, and implications of the study. Moreover, in light of the fact that the participants of this study are Spanish-speaking monolinguals, the scope of the title “Gestures as Scaffolding to Learn Vocabulary in a Foreign Language” could be adjusted. Furthermore, since the first sentence of the abstract indicates that the study is a review, it may be confusing to the reader as it also contains empirical research. There are over 170 references cited by the authors to make a comprehensive review of the topic, but whether it is a qualitative or a review is unclear. Therefore, the authors should revise the manuscript and include clear research gaps in order to make it more precise.

Author Response

Reviewer 2.

Dear authors, thank you for conducting such a comprehensive study. I have the following concerns:

  1. Firstly, in the Abstract, there is too much detailed information about this study. It is suggested that the Abstract be concise and focus on the rationale, methodology, results, and implications of the study.

We would like to thank to the reviewer for this comment. Following this suggestion, we have reviewed the abstract, removing some details that are explored more in-depth throughout the manuscript (please, see p. 1). Now, after introducing our work, each of the experiments is presented along with the results, and a final conclusion about the experimental outcomes is provided.

  1. Moreover, in light of the fact that the participants of this study are Spanish-speaking monolinguals, the scope of the title “Gestures as Scaffolding to Learn Vocabulary in a Foreign Language” could be adjusted.

Indeed, we have had a similar discussion during the revisions of our previous published papers on the topic. As clarified in the participants section, our intent was to include individuals in our research who were as low proficient as possible in any foreign language. In our first study on the topic, published in 2019, we consistently referred to the foreign language as the second language, and participants were labeled as “Spanish monolingual speakers”. However, there is a noticeable change in our second study (2021). One of the reviewers suggested that university students, being exposed to language learning during formal education, cannot be accurately called monolingual speakers. In response to this suggestion, we used the term “monolinguals” and provided relevant information about inclusion and exclusion criteria. Aligning with this recommendation, instead of using the term “second language”, the reviewer proposed the use of “foreign language”, which is also a prevailing trend in the field. In this way, we thought that the title of this manuscript is all-encompassing and embraces many of the real-life situations in which people acquire a new language beyond their native tongue.

  1. Furthermore, since the first sentence of the abstract indicates that the study is a review, it may be confusing to the reader as it also contains empirical research. There are over 170 references cited by the authors to make a comprehensive review of the topic, but whether it is a qualitative or a review is unclear. Therefore, the authors should revise the manuscript and include clear research gaps in order to make it more precise.

Thank you very much for your comment. We agree with the reviewer that the previous version of the manuscript did not clarify the type of article in question (empirical paper, review article, etc.). Specifically, the current manuscript is a theoretical review of the results obtained in three experiments reported in two already published articles:

- García-Gámez, A. B., & Macizo, P. (2019). Learning nouns and verbs in a foreign language: The role of gestures. Applied Psycholinguistics, 40(2), 473-507. https://doi.org/10.1017/S0142716418000656  

- García-Gámez, A. B., Cervilla, Ó., Casado, A., & Macizo, P. (2021). Seeing or acting? The effect of performing gestures on foreign language vocabulary learning. Language Teaching Research, 1-32. https://doi.org/10.1177/13621688211024364

The aim of this manuscript is to provide a unified theoretical integration of the findings obtained in these previous studies. This goal could not be accomplished in the discussions of each article as they were published independently as the experimental series were conducted and we were learning from the findings obtained in each of them.

Following the reviewer's indications, the goals of this paper have been clearly detailed in the revised version of the manuscript, abstract (p. 1), introduction section (p. 8), title of sections 2 and 3 of the manuscript (p. 9 and p. 17, respectively) and the discussion section (p. 20).

We would also like to thank the reviewer for the methodological clarifications recommended. In response to this and to methodological comments raised by reviewer 1, in the revised version of the manuscript, we have clarified key issues such as the form of presentation of the learning material (blocked) the timing of evaluation (concurrent), etc. (please, see pp. 11-32 for Experiments 1 and 2, and p. 18 for Experiment 3).

We thank reviewer 2 for the detailed reading of our work and for the indications aimed at improving the scientific quality of the manuscript.

Round 2

Reviewer 2 Report

Comments and Suggestions for Authors

Dear authors, thank you for revising your manuscript according to the suggestions from the reviewers. After your careful revision, it appeared to be much more coherent and rigorous. The last concern I have is to suggest you shorten the introduction section to make the rationale of conducting this study more concise.

Author Response

Please, see attached file.
